# The Role of Infant Formulas in the Primary Prevention of Allergies in Non-Breastfed Infants at Risk of Developing Allergies—Recommendations from a Multidisciplinary Group of Experts

**DOI:** 10.3390/nu14194016

**Published:** 2022-09-27

**Authors:** Jorge Amil Dias, Edmundo Santos, Inês Asseiceira, Sylvia Jacob, Carmen Ribes Koninckx

**Affiliations:** 1Paediatric Gastroenterology, Hospital Lusíadas, 4050-115 Porto, Portugal; 2Neonatology Unit, Hospital de São Francisco Xavier—Centro Hospitalar de Lisboa Ocidental, 1349-019 Lisboa, Portugal; 3Laboratório de Nutrição, Faculdade de Medicina de Lisboa, Centro Hospitalar Universitário Lisboa Norte, 1649-028 Lisbon, Portugal; 4Paediatric Allergology Unit, Centro Hospitalar Universitário São João, 4200-319 Porto, Portugal; 5Paediatric Gastroenterology & Hepatology & Instituto de Investigacion Sanitaria, Hospital Universitario y Politecnico La Fe, 46026 Valencia, Spain

**Keywords:** primary prevention of allergy, hydrolysed formulas, synbiotics, prebiotics, probiotics

## Abstract

The worldwide incidence of allergic diseases has been continuously increasing, and up to one in every five people are currently affected by these medical conditions. Although seldom fatal, allergies have a profound impact on children’s growth, development, and quality of life, besides being associated with heavy healthcare costs and resource utilisation. In this context, a group of experts in nutrition, paediatric gastroenterology, allergology, and neonatology joined forces to discuss the role of infant formulas in the primary prevention of allergies in infants for whom breastfeeding is not an option and who are at risk of developing allergies. The topics discussed included the assessment of risk, the impact of the microbiota on the modulation of immune tolerance, and the added value of certain formula characteristics, namely, protein integrity (hydrolysed protein vs. intact protein) and the addition of prebiotics, probiotics, or synbiotics. This article describes the latest evidence on each of the above-mentioned points, as well as a number of recommendations made by the experts to guide counselling of parents in the choice of a formula for infants at risk of allergy. Overall, the experts highlighted family history and dysbiosis-promoting factors (namely, caesarean delivery and antibiotic use) as two of the most important risk factors for allergy development. Moreover, in line with international guidelines, the panel advocated that intact protein formula should be offered to all bottle-fed healthy infants, irrespective of their allergic risk (with the exception of short-term bottle feeding of otherwise breastfed babies in their first week of life, for whom a hydrolysed formula may be advisable). Finally, the experts agreed that the use of prebiotic-, probiotic-, or synbiotic-enriched formulas should be considered in infants at risk of developing allergies.

## 1. Introduction

Allergies are a global public health issue that affect approximately 20% of the worldwide population, being particularly prevalent among people living in industrialised and westernised regions [1]. Food allergies, atopic dermatitis, and asthma are common childhood allergies and key steps of the atopic march, which is a temporal pattern of allergy development that affects the skin, the gastrointestinal tract, and the respiratory tract in a time-based order [1]. Being one of the initial steps of this march, food allergies are characterised by a disproportionate immune reaction to specific foods that, ultimately, can lead to an anaphylactic reaction [2,3]. Up to 10% of the worldwide population is affected by food allergies; among those, cow’s milk allergy (CMA) is one of the most common and affects 2% to 3% of all infants [2,3,4,5]. Atopic dermatitis is a frequent skin inflammatory disorder with varying activity and evolution patterns that affects up to 20% of children and 10% of adults in high-income countries [6]. Finally, asthma is the most common chronic respiratory disorder; while its prevalence varies widely across the globe, it seems to be particularly high in English-speaking countries, where it can affect 8% to 12% of the entire population [7]. Globally, the prevalence of allergies has been rising over the last decades; interestingly, however, not all the above-mentioned disorders evolve equally across the globe. In fact, the prevalence of both atopic dermatitis and asthma seems to have reached a plateau in westernised regions, while an increasing trend prevails among low- to middle-income countries [6,7,8,9]. Importantly, all discussed allergies represent a heavy societal burden: not only do they significantly impact the quality of life of those affected and, in some cases, may actually have a long-term influence on children’s growth and development but they are also associated with substantial direct and indirect costs [2,5,6,9]. The investment in effective preventive strategies for those at risk is, therefore, a healthcare priority.

Breastfeeding is unquestionably the best feeding method and the most complete source of nutrients for babies during their first months: the World Health Organization (WHO) and the United Nations Children’s Fund recommend breastfeeding exclusively for the initial 6 months and continuously for up to 2 years or beyond [10]. However, not all mothers are able or willing to do so; in such cases, infant formulas are the only suitable substitute for human milk. In fact, infant formulas have been developed to resemble human milk as closely as possible; although they are commonly based on cow’s milk, its contents are diluted and modified in order to approach the nutrient distribution found in breast milk [11]. This article is based on a series of advisory board meetings that were held with the following goals: (i) to discuss the role of infant formulas in the primary prevention of allergies in non-breastfed babies who are at risk of developing allergies; and (ii) to issue a number of recommendations aimed at bringing clinical practice closer to scientific evidence. The advisory boards’ panel was composed of experts in nutrition, paediatric gastroenterology, allergology, and neonatology—most of whom authored this article—and the meetings took place in December 2021 and April 2022. The main topics discussed during these meetings are briefly reviewed in the next sections.

## 2. Risk Factors for Infant Allergies

Any effective preventive strategy must rely on accurate identification of the individuals at risk; in fact, primary prevention approaches often consist of the modulation of one (or more) modifiable risk factors. As in many other medical conditions, allergies’ risk factors are multifactorial and involve genetic traits, environmental aspects, and a complex pattern of genome-environment interactions. Those most commonly related to food allergies, atopic dermatitis and asthma are summarised in Figure 1 [1,3,5,6,7]. Some of these risk factors are immutable, as is the case of familial associations, race/ethnicity, and, to some extent, skin barrier dysfunction; others, however, can be at least partially modulated as part of a preventive approach. The latter include vitamin D insufficiency, diet (including timing and route of exposure to certain food allergens), exposure to air pollution and tobacco smoke, certain aspects related to the dysfunction of the skin barrier (such as the physical effect elicited by scratching), and the composition of the epidermal, respiratory, and gastrointestinal microbiota. The family history and the composition of the microbiota were deemed as two of the most important risk factors by the scientific panel present on the advisory boards; the former is readily identifiable and has a considerable weight in the development of allergies, whereas the latter is increasingly acknowledged as one of the key aspects that mediates the acquisition of immune tolerance, as well as a potential target of several preventive strategies. As such, these risk factors will be further explained in the next sections.

### 2.1. Family History of Atopic Disease

Among the identifiable allergy-related risk factors, a family history of atopic disease is probably one of the strongest. Although a considerable number of infants without known familial associations do develop allergies, the incidence of these medical conditions is much higher among those who have direct relatives affected. In fact, the heritability of CMA has been estimated to be 15%, whereas that of asthma is between 35% to 95% [5,7]. Moreover, 66.6% and 13.6% of children with at least one sibling affected by food allergies were shown to be food-sensitised and clinically reactive, respectively [3]. Additionally, studies with twins have shown heritability as high as 75% for atopic dermatitis and 82% for asthma [6,7].

This strong genetic predisposition relies on an increasingly known set of genes, which encode proteins that are usually involved in the modulation of immune responses (including—but not limited to—innate immunity, type 2 T cell differentiation and T cell activation), and in the differentiation and development of the epidermis. For instance, *HLA* (human leukocyte antigen) genes, which encode cell-surface proteins involved in the regulation of the immune system, are associated with the development of several allergic diseases (e.g., *HLA-DR* and *HLA-DQ* regions are associated with peanut allergy, and *HLA g* is associated with asthma) [3,7]. Moreover, a type 2 cytokine cluster (which codifies two well-known cytokines, interleukin (IL)-4 and IL-13, and a cytokine expression switch, RAD50) is associated with the development of atopic dermatitis [6]. Finally, mutations in the *FLG* gene—which encodes filaggrin, a structural protein with an important role in the skin’s barrier—is the strongest genetic risk factor associated with atopic dermatitis; in fact, 20% to 40% of all patients with atopic dermatitis have *FLG* loss-of-function mutations [6]. Interestingly, these mutations are also related to the development of asthma and peanut allergy, demonstrating the importance of skin barrier integrity in allergies other than atopic dermatitis [6].

### 2.2. Dysregulation of the Microbiota

The human microbiota is a complex microbial ecosystem composed of commensal, symbiotic, and pathogenic microorganisms that can be found in the gut, skin, oral cavity, nasal passages, and urogenital tract [12,13]. Of these, the bacterial gut microbiota is the most well-studied and has been acknowledged as a key factor in the development of a healthy and resilient immune system. The colonisation of the gut is a dynamic process that is thought to begin at the foetal stage, progressing through an ecologically ordered succession of species until reaching a steady and balanced composition (which occurs approximately 1000 days after birth) [12,13,14,15,16]. As a consequence, the number and diversity of gut-associated bacterial species depend on a series of pre-, peri-, and post-natal factors (Figure 1). In fact, and notwithstanding a substantial intra- and interindividual variation, the microbiota of vaginally delivered babies differs significantly from that of caesarean-born ones, who are deprived of exposure to maternal vaginal and faecal microbiota [12,13,14,15,16,17]. Likewise, breast milk is a source of colonising microbes and of a number of active compounds—such as human milk oligosaccharides (HMOs)—which promote the growth and expansion of beneficial bacterial strains in the gut, namely *Bifidobacterium* spp. Formula-fed babies are deprived of both the milk-colonising microbes and the HMOs and tend to present a more diverse and *Clostridium*-dominated gut microbiota [12,13,14,15,16,17]. Other factors that may impact the composition of infants’ microbiota are genetics, the use of antibiotics (either during pregnancy, within delivery–particularly common in caesarean deliveries–and/or after birth), the use of other drugs, pregnancy health and gestational age, and early environmental exposures (namely, to siblings or household pets, and rural-like contexts) [12,13,14,15,16,17].

The composition of the gut microbiota has a key role in organisms’ protection and immune system development. In fact, not only do the gut bacterial species inhibit the growth of pathogenic bacteria through nutrient competition and production of antimicrobial peptides and toxins but they also promote the acquisition of immune tolerance (i.e., the non-responsiveness to self and harmless environmental antigens) [12,13,14,17,18,19,20]. This promotion of immune tolerance is achieved through the suppression of the T helper type 2 (Th2) response, which has a dominant role in the pro-allergic processes, and through the preservation of the intestinal epithelial barrier integrity, which averts the access of ingested allergens to the systemic circulation. The microbiota impact these processes mainly by promoting the expansion and the activity of regulatory T cells (Treg cells), responsible for suppressing the Th2 phenotype, and by inducing the secretion of immunoglobulin A (IgA), which helps seal the intestinal epithelial barrier [13,14,18,19,20]. As a consequence, it is rational to assume that the occurrence of dysbiosis (i.e., an unbalanced composition of the microbiota) during infancy is favourable to the development of allergies and other immune-mediated conditions. Additionally, indeed, several research groups have repeatedly demonstrated that the composition and abundance of gut microbiota in children with allergies differ from that found in healthy controls [14,18,20,21,22,23,24,25]. Moreover, the presence or absence of certain bacterial species in babies’ gut microbiota has been associated with an increased risk of atopy later in life, especially the development of asthma, eczematous diseases, and several food allergies [13,14,18,24,25,26]. Additionally, the timing of host–microbe interactions were shown to be important, as a particular study on children with CMA demonstrated that the gut microbiota composition between the 3rd and the 6th month of age—but not afterwards—was associated with allergy resolution when the subjects were 8 years old [17,18,23,24].

The observational studies in human cohorts depicted above were fundamental to demonstrating an association between microbiota composition and allergy development but were unable to show causality. Mechanistic studies on mice have, therefore, been essential to shed some light on the key aspects mediating dysbiosis and an atopic-prone state. In fact, germ-free mice were shown to be predisposed to several allergies, while presenting a marked bias towards Th2 responses and a reduced population of Treg cells [13,17,23,24,26,27]. Of note, the colonisation of these mice (or of antibiotic-treated mice) with commensals from healthy infants protected against allergic symptoms; conversely, the transference of microbiota from allergic children failed to mitigate the allergic reactions [13,18,21,23,24,26]. Moreover, the administration of Clostridiales and Bacteroidales to gnotobiotic mice (i.e., mice with defined microbial communities) induced the expansion of Treg cells and the production of IgA (among other immunological-related effects), repairing the intestinal barrier protective function and ultimately averting the baseline allergic-prone phenotype [13,17,18,21,24,27].

## 3. The Role of Infant Formulas in Allergy Prevention

As previously noted, breastfeeding is universally acknowledged as the optimal feeding method for babies up to 6 months of age. However, when exclusive breastfeeding is not an option, infant formulas are used as a complement or as a replacement for human milk. Standard infant formulas are usually based on cow’s milk and manufactured in order to be as close as possible to human milk. According to the EFSA (European Food Safety Authority), these formulas should have 60 to 70 kcal/100 mL, and a total of 1.8 to 2.5 g of protein, 4.4 to 6.0 g of fat, and 9.0 to 14.0 g of carbohydrates per 100 kcal (besides a number of other essential elements) [28]. In addition to these standard formulas, the industry has developed a number of specialised formulas particularly shaped to meet the needs of newborns and infants who present specific disorders or medical conditions (such as reflux, CMA, lactose intolerance, or prematurity). These formulas differ from the standard ones in several characteristics, namely the integrity of the cow’s milk proteins, the whey:casein ratio, the percentage of lactose, the amount of protein, and/or carbohydrates, and the presence of specific supplements (e.g., minerals, vitamins, and prebiotics or probiotics, among others). As such, and considering the increasingly high incidence of allergies, one may wonder whether any of these characteristics may prevent or ameliorate the development of atopic diseases. In this section, we will review the evidence concerning two main formula-differentiating features—the protein integrity and the addition of prebiotics, probiotics, or synbiotics—in the primary prevention of allergies in partially or totally bottle-fed babies.

### 3.1. Partially and Extensively Hydrolysed Formulas vs. Intact Protein Formulas

Partially hydrolysed and extensively hydrolysed formulas (pHF and eHF, respectively) are two types of infant formulas characterised by the use of hydrolysed cow’s milk proteins. Accordingly, these formulas were initially developed for newborns or infants showing signs of intolerance to cow’s milk protein. Although there is no formal definition for pHF and eHF in terms of peptide size, this usually falls between 3 and 10 kDa (usually <5 KDa) in the former and is less than 3 KDa in the latter [29]. Based on a series of studies—further discussed below—pHF and eHF were once considered to be hypoallergenic formulas and their use became popular as a primary prevention strategy in infants considered to be at risk of allergies. Of note, while the worldwide average rate of utilisation of pHF is approximately 7%, this value varies widely across geographical regions and can be as low as 0% or 1% (in Spain and Italy, respectively) or as high as 47% (in Portugal).

The rationale for using pHF and eHF as a preventive measure against allergies is based on the reduced allergenicity of small peptides when compared to intact proteins, decreasing the risk of sensitisation and promoting the induction of oral tolerance in infants at risk [29]. In fact, a number of initial studies supported this theory by demonstrating an association between the utilisation of pHF and eHF and a reduction in the incidence of atopic diseases [30,31]. As a consequence, international societies have issued recommendations accordingly, although such recommendations were formulated in a somewhat cautious manner. Some years ago, the American Academy of Pediatrics (AAP) claimed that infants at high risk “may benefit from exclusively breastfeeding or a hypoallergenic formula or possibly a partial hydrolysate formula” [32], whereas the American Academy of Allergy, Asthma, and Immunology stated that “for infants at increased risk of allergic disease who cannot be exclusively breast-fed for the first 4 to 6 months of life, hydrolysed formula appears to offer advantages to prevent allergic disease and cow’s milk allergy” [33]. Moreover, a joint statement from the European Society for Paediatric Allergology and Clinical Immunology and from the European Society for Paediatric Gastroenterology, Hepatology, and Nutrition advocated the “exclusive feeding of a formula with a confirmed reduced allergenicity” in bottle-fed infants with a documented hereditary atopy risk [34]. Finally, the European Academy of Allergy and Clinical Immunology (EAACI) claimed that there is significant evidence regarding the benefits of hydrolysed formula and recommended the use of a “hypoallergenic formula with a documented preventive effect” in high-risk infants during the first 4 months when breastfeeding is insufficient or not possible (evidence level I, grade A-B) [35].

The above-mentioned recommendations were issued 8 to 23 years ago (as will be further discussed below); meanwhile, new studies were conducted, and new evidence has been gathered. The latest assessments on the subject—which include a systematic review and meta-analysis including over 19,000 participants—found no consistent evidence supporting a role for pHF and eHF in the primary prevention of children’s allergies [36,37,38,39]. The reason for this conflicting evidence is unclear; however, it has been suggested that the qualitative changes suffered by the peptides and the hydrolysis method—rather than just the degree of hydrolysis—impact the preventive potential of each formula [29]. As a consequence, findings for a particular formula should not be generalised to other formulas. Accordingly, Vandenplas et al. claimed that meta-analyses focused on trials with a similar design, and a single hydrolysate did demonstrate a limited effect on allergy prevention, particularly when it comes to the development of atopic dermatitis [40]. Still, Boyle et al. found evidence of publication and methodological biases in several studies reporting allergic outcomes after the use of hydrolysed formulas [36]. Therefore, the current position assumed by most authors is that the available evidence is insufficient to demonstrate a substantial benefit of pHF and eHF in the prevention of allergies in infants at risk [36,37,38,39,40]. A similar conclusion was reached by Osborn et al. in the most recent Cochrane review on the topic: evidence from 16 studies suggested that the prolonged use of hydrolysed infant formulas had no impact on the development of childhood allergic disease, asthma, eczema, rhinitis, food allergy, or CMA when compared to standard cow’s milk formula (based on very low-quality evidence) [41]. The only exception concerns short-term feeding in otherwise breastfed babies (for instance, during the 3 to 4 days after delivery); in these cases, the authors concluded that the use of an eHF may prevent CMA (although also based on very low-quality evidence) [41]. In fact, a recent randomised trial has shown that the incidence of CMA at 24 months of age was significantly lower among babies who were breastfed with or without supplementation with an amino-acid elemental formula (AAF) for at least the first 3 days of life when compared to those who were breastfed and supplemented with an intact protein formula from the first day of life to up to their fifth month [42].

The evidence presented above suggests that infant formula protein hydrolysis has no role in preventing the development of allergic diseases. In addition, a recent observational study has demonstrated that the use of pHF at 2 months of age is associated with a higher risk of wheezing, food allergy, and, to a lesser extent, eczema in babies up to the age of 2 years [43]. In this scenario, one may wonder whether the ingestion of intact cow’s milk formula (vs. pHF and eHF) early in life may actually promote oral tolerance to cow’s milk. Indeed, a number of studies have shown that the early (up to 3 months) introduction of intact cow’s milk formula is associated with a reduced risk of CMA, even when the risk estimation is adjusted for confounding variables (such as atopic familial history) [37,44,45,46]. Of note, the mentioned studies are observational, and rigorously designed clinical studies are needed to fully explore this association and to demonstrate causality. Still, the introduction of allergenic foods between 4 and 7 months has been generally acknowledged and recommended by international guidelines as a strategy to prevent the development of food allergies.

The formulas’ overall resemblance to human milk—considered to be the gold standard in babies’ nutrition—is another point that should be considered in this discussion. In fact, standard formulas are manufactured to resemble breast milk: to achieve such a goal, bovine milk is diluted and skimmed so that all components fall within a range of values similar to those found in breast milk [11,47]. Specialised infant formulas, however, diverge from that gold standard in a few specific parameters (considered to be safer or more appropriate for infants with medical or dietary issues) [11,47]. Accordingly, intact protein formulas are closer to human milk than partially hydrolysed and extensively-hydrolysed ones, as proteins in human milk are generally in their intact (i.e., non-hydrolysed) form. Additionally, it should also be noted that intact protein formulas have better palatability than hydrolysed ones [11].

### 3.2. Synbiotics-Enriched Formulas

As previously noted, infant formulas are manufactured in such a way as to resemble breast milk as closely as possible. Importantly, the composition of breast milk goes beyond nourishment; besides fat, proteins, and carbohydrates, breast milk includes a diverse array of non-nutritive components such as HMOs, bacterial strains, antibodies, hormones, and immune cells. In an attempt to mimic this highly functional cocktail, manufacturers have been exploring the possibility of enriching infant formulas with a vast selection of ingredients besides those commonly found in bovine milk. Prebiotics, probiotics, and synbiotics (a combination of the former two) are among such ingredients.

The International Scientific Association for Probiotics and Prebiotics defines prebiotics as a “substrate that is selectively utilized by host microorganisms conferring a health benefit” [48]. Examples of prebiotics currently used in infant formulas include oligosaccharides (such as galacto-oligosaccharides [GOS], fructo-oligosaccharides [FOS], 2′-fucosyllactose, and lacto-N-neotetraose), soluble fermentable fibres, and plant polyphenols [12,48]. Probiotics, on the other hand, have been described by the WHO and the Food and Agriculture Organization as “live organisms which, when consumed in adequate amounts, confer a health benefit to the host” [49]. *Bifidobacterium* and *Lactobacillus* are two of the most commonly used genera in this context [12]. The health benefits provided by both prebiotics and probiotics are thought to result from a complex interplay between those elements, the host cells, and the host resident microorganisms [50]. Overall, prebiotics and probiotics are believed to have an important role in the modulation of the gut microbiota (through the promotion of the expansion of beneficial bacterial populations and the competitive exclusion of pathogenic ones) and in the regulation of the immune system (by impacting mucosal immune mechanisms, regulating the secretion of immunoglobulins and cytokines, and supporting the integrity of the intestinal epithelial barrier) [12,49,50]. As such, and given the conspicuous crosstalk linking the microbiota, the immune system, and the development of allergies (reviewed above), one may argue that prebiotics and probiotics are likely to play a role in this dynamic and aid in the suppression of atopic phenotypes. The evidence concerning such a hypothesis is summarised in the next paragraphs.

The outcomes of several preclinical and in vitro studies involving prebiotics and/or probiotics do support a role for these elements in the prevention of allergic diseases, particularly through the maturation of the intestinal epithelial barrier and the regulation of the immune system [4,14,17,27]. The impact on the immune system is mostly mediated by the establishment of an appropriate Th1/Th2 response balance, the development and expansion of Treg cells, an increase in the secretion of anti-inflammatory cytokines, and the suppression of the IgE production, thereby promoting immune tolerance and reducing allergic symptoms [4,14,17,27]. In fact, a mixture of oligosaccharides (including short-chain GOS [scGOS] and long-chain FOS [lcFOS]) was shown to induce immunological tolerance and suppress the allergic phenotype in whey-sensitised mice [4]. Moreover, the probiotic administration of different bacterial strains (namely *Bifidobacterium infantis* and *Bifidobacterium breve* M-16V) to experimental models of food allergy induced visible changes in the resident microbiota and/or substantially reduced atopic manifestations [14,17]. Finally, the use of synbiotic mixtures in mice models was also associated with promising outcomes: the administration of *B. breve* M-16V and GOS/FOS protected against the development of CMA; the administration of *B. breve* M-16V, β-lactoglobulin-derived peptides, and scFOS/lcFOS suppressed an acute allergic skin response; and the administration of *B. breve* M-16V and non-digestible oligosaccharides induced a Treg response and reduced pulmonary airway inflammation and remodelling in an asthma animal model [14].

The above-described results, obtained in the preclinical context, were supported and validated by a number of observational or interventional clinical trials. In fact, supplementation with prebiotics was shown to have beneficial effects: in particular, the development of atopic dermatitis in high-risk infants was suppressed upon supplementation with a formula containing a particular mixture of oligosaccharides (scGOS/lcFOS, 9:1 ratio), which also seemed to promote a bifidogenic environment in the gut (closer to that observed in vaginally born and breastfed babies) [4,26]. Results with probiotics are known to be highly strain-specific, and some strains have been associated with particularly promising results. For instance, a number of studies in children with CMA and/or eczema have shown that the addition of *Lactobacillus rhamnosus* GG (LGG) to an eHF promoted changes in the microbiota, accelerated immune tolerance acquisition and allergy resolution, led to a lower incidence of other atopic manifestations, and decreased inflammation and other gastrointestinal symptoms (when compared with eHF alone) [4,24,27]. *B. breve* M-16V was shown to modulate the microbiota and reduce the severity of allergic symptoms in bifidobacteria-deficient infants, as well as to increase the levels of short-chain fatty acids (known to be related to the *Bifidobacterium* spp. abundance) and Treg-secreted chemical signals in pre-term infants [14]. Additionally, the administration of three *Bifidobacterium* spp. strains improved the symptomatology and increased the quality of life of children with allergic rhinitis [14]. Of note, many of the clinical trials carried out in this area are not focused on a specific prebiotic or on a particular strain of probiotics, but rather explore the synergistic and complementary effects of both (i.e., of synbiotics). For example, the addition of a prebiotic blend including FOS and *B. breve* M-16V to an AAF was shown to increase *Bifidobacterium* spp. counts in infants with suspected non-IgE CMA, reaching values similar to those seen in breastfed infants [4,15]. Similarly, the gut physiological environment of caesarean-born children was shown to approach that of vaginally born ones upon supplementation with *B. breve* M-16V and scGOS/lcFOS [40]. Moreover, the administration of *B. breve* M-16V and scGOS/lcFOS to 5-month-old babies with atopic dermatitis was associated with relief from allergic symptoms (although only in those who had elevated IgE levels); additionally, a lower prevalence of asthma-like symptoms was observed in the whole experimental group when the subjects were aged 1 year [15,26].

## 4. International Guidelines and Regulatory Authorities

The scientific evidence pertaining to allergy development is routinely reviewed by several international bodies that issue recommendations in the field, aiming to guide paediatricians and other healthcare professionals during parents’ counselling. Table 1 summarises the main recommendations concerning infant formulas in the context of allergy prevention made by the EAACI, the Australasian Society of Clinical Immunology and Allergy, the AAP, the EFSA, and the World Allergy Organization (WAO) [51,52,53,54,55,56].

As previously noted, pHF and eHF were once recommended as an effective prevention measure against allergies; however, in view of the latest evidence, such recommendations have been replaced by a general consensus that standard cow’s milk formula can be used in all bottle-fed infants irrespective of their allergy risk [57]. In fact, the international entities listed on Table 1 acknowledge that there is no evidence supporting a preventive role for pHF or eHF in the development of allergies and, therefore, no longer recommend these alternatives for high-risk infants [51,52,53,54]. It should be noted, however, that the level of evidence for such recommendations is low to moderate, suggesting the need for more studies on this subject. Moreover, the EAACI recommends against supplementation with standard cow’s milk formula during the first week of life in breastfed infants, claiming that such action may increase the risk of CMA (although the level of evidence is once again considered to be low) [51]. The EAACI committee further considers that breastfeeding is usually sufficient in healthy infants; however, should it become necessary to supplement during this initial week, donor breastmilk, hydrolysed formula, AAF, or water may be suitable options (depending on clinical, cultural, and economic factors) [51]. Reference to the use of prebiotics, probiotics and synbiotics was made by two of the entities listed in Table 1: whilst both acknowledge that the supporting evidence is of low or very low certainty, the EAACI made no recommendation (for or against), while the WAO panel considered that prebiotic and probiotic supplementation might be beneficial in high-risk bottle-fed infants [51,55,56].

## 5. Expert Comment

Based on the evidence discussed above and our own personal expertise, we hereby present several recommendations concerning the use of formulas and their role in preventing allergies in infants at risk of developing allergies (Box 1). First and foremost, we do believe that exclusive breastfeeding is the best and most complete option to nourish babies and promote their adequate development until their sixth month of age. Breastfeeding should therefore be duly encouraged and promoted by healthcare professionals, and breastfeeding support should be offered to all parents who need it. However, when a mother is unable or unwilling to breastfeed, infant formulas should be recommended as the only suitable replacement for human milk. We have reviewed the available evidence concerning the use of infant formulas in allergy-prone babies from two different perspectives: protein integrity (pHF or eHF vs. intact protein formulas) and the addition of prebiotics, probiotics, or synbiotics.

Paediatricians and general practitioners attending to young babies should carefully assess their allergy risk; whereas some risk factors are not readily discernible, others can be objectively observed and/or evaluated. We consider that infants with a familial history of atopy (i.e., who have parents or siblings with allergic manifestations) and/or those who have dysbiosis-promoting factors (namely pre-, peri-, or post-partum antibiotic exposure and/or who were born by caesarean delivery) should be considered at increased risk of developing allergies.

When an infant is considered to be at risk of developing allergies and breastfeeding is not a possibility, medical doctors and nurses should carefully consider the available formulas and advise the parents on the best solution for their child. Regarding protein integrity, we consider that the available evidence does not support the once-thought preventive role attributed to pHF and eHF in allergy development. In addition, an early offer of intact protein (beyond the age of 4 months) is consistent with the current trend of introducing food allergens as early as possible. Moreover, intact protein formulas are closer to the gold standard (i.e., human milk) than pHF or eHF, as well as more palatable and usually less expensive. For those reasons, we recommend the use of intact protein formulas in all bottle-fed infants, irrespective of their allergy risk. The only exception to this recommendation is the short-term bottle feeding of otherwise breastfed babies in their first week of life (for instance, during the 3 to 4 days that follow delivery while both mother and baby are still hospitalised). In such cases, supplementation with intact protein seems to aggravate the risk of CMA. Therefore, and when needed, the use of other supplementary options—such as donor breast milk, eHF, or AAF—may be advisable, although it should be acknowledged that the latter two options are expensive and have little evidence to support their use.

Regarding the role played by prebiotics, probiotics, and synbiotics in allergy prevention, we do acknowledge that the evidence comes mostly from preclinical and observational studies, and therefore causality is yet to be demonstrated. However, studies made on a few specific components—such as LGG, *B. breve* M-16v and certain oligosaccharides—do demonstrate an important effect on microbiota modulation and immune tolerance acquisition. Further, to the best of our knowledge, no adverse events have been associated with the use of prebiotics and/or probiotics. Therefore, we do believe that the use of probiotic-, prebiotic-, or synbiotic-enriched formulas may be beneficial to bottle-fed infants with a risk of developing allergies and that this option should be considered and discussed with the parents.

Importantly, there are still major knowledge gaps in the effect of nutrition on allergy prevention: all recommendations described above are based on limited or very limited evidence, and further studies are of utmost importance to confirm (or contradict) our views. Given the continuously increasing allergy incidence and its consequences in terms of child development, growth, and quality of life, a huge effort should be placed on the identification of effective preventive mechanisms. We, therefore, urge infant formula manufacturers and the scientific community to gather together their efforts in order to fully comprehend the mechanisms underlying allergy development, so that effective nutritional strategies can be established (particularly during the critical first 1000 days).

A final note goes to the importance of healthcare professionals adapting their recommendations to the latest scientific evidence. We believe that paediatricians, general practitioners and other healthcare personnel attending young babies have a responsibility to advise parents on how to best feed their infants, according to each specific and potential health issue. Even though pHF and eHF are no longer recommended as allergic-preventive measures, many medical doctors and nurses still advocate their use in high-risk infants (as seen by the rate of the utilisation of these formulas in some countries). We would therefore like to make a public call-to-action to all healthcare professionals to periodically review the scientific literature and the international recommendations, adapting their practice accordingly.

Box 1Summary of the Panel’s Recommendations.
-Allergy-related familial history and the presence of dysbiosis-promoting factors (namely pre-, peri-, or post-partum antibiotic exposure and caesarean delivery) should be evaluated during the infant’s initial medical appointments in order to ascertain the risk of allergy.-When exclusive breastfeeding is not an option, intact protein cow’s milk-based formula should be used to feed or supplement healthy infants, irrespective of their allergy risk *.-Without disregarding the previous point, the short-term bottle feeding of otherwise breastfed babies up to 1 week of age should be avoided (as breastmilk is usually sufficient); however, and when needed, adequate supplementary options include donor breast milk, eHF, or AAF, although it should be acknowledged that these options are expensive *.-A probiotic-, prebiotic-, or synbiotic-enriched formula should be considered in non-breastfed infants who are at risk of developing allergies *.
* Based on limited evidence; eHF, extensively hydrolysed formula; AAF, amino acid-based formula.

## Figures and Tables

**Figure 1 nutrients-14-04016-f001:**
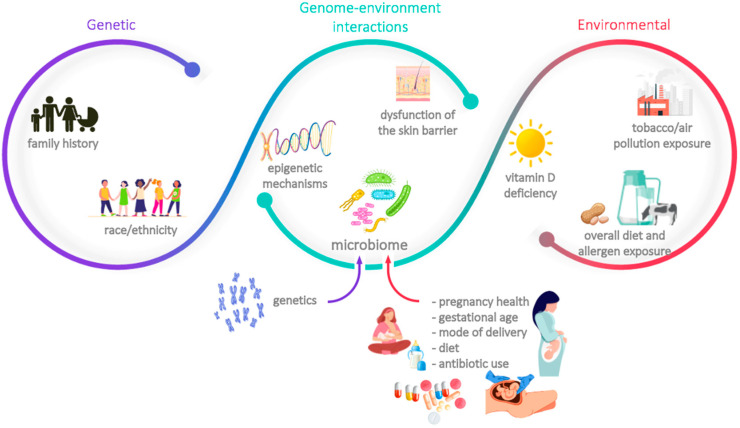
Main risk factors associated with the development of allergies.

**Table 1 nutrients-14-04016-t001:** Recommendations made by paediatrics and gastroenterology international societies and regulatory entities in the context of primary allergy prevention.

International Society/ Regulatory Entity	Recommendations Concerning *:	
	Hydrolysed Formulas vs. Intact Protein Cow’s Milk Formulas	Addition of Prebiotics, Probiotics, and Synbiotics to Infant Formulas
EAACI 2020 [51]	- No recommendation for or against pHF or eHF to prevent food allergies in infants. - **Level of evidence: low** - Avoid supplementation with cow’s milk-based formula in breastfed infants in the first week of life to prevent CMA. - **Level of evidence: low**	- No recommendation for or against the use of prebiotics, probiotics or synbiotics in infants to prevent food allergies. - **Level of evidence: low**
ASCIA 2019 [52]	- When breastfeeding is not possible, a standard cow’s milk-based formula can be given; pHF or eHF should not be given for primary prevention of allergies. - **Level of evidence: moderate**	-
AAP 2019 [53]	- There is no evidence that pHF or eHF prevents atopic disease in infants and children, even in those at high risk for allergic disease.	-
EFSA 2021 [54]	- No conclusions could be drawn on the efficacy of a particular formula (containing a specific protein hydrolysate derived from whey protein isolate) in reducing the risk of developing atopic dermatitis in infants with a family history of allergy.	-
WAO 2015 [56]	-	- The WAO guideline panel suggests using probiotics in infants at high risk of developing allergies because, considering all critical outcomes, there is a net benefit resulting primarily from the prevention of eczema. - **Conditional recommendation** - **Level of evidence: very low**
WAO 2016 [55]	-	- The WAO guideline panel suggests prebiotic supplementation in non-exclusively breastfed infants, both at high and at low risk for developing allergy. - **Conditional recommendation** - **Level of evidence: very low**

* Level of evidence as reported; AAP, American Academy of Pediatrics; ASCIA, The Australasian Society of Clinical Immunology and Allergy; CMA, cow’s milk allergy; EAACI, European Academy of Allergy and Clinical Immunology; EFSA, European Food Safety Authority; eHF, extensively hydrolysed formula; pHF, partially hydrolysed formula; WAO, World Allergy Organization.

## Data Availability

Not applicable.

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
