# Peer review of "The Role of Infant Formulas in the Primary Prevention of Allergies in Non-Breastfed Infants at Risk of Developing Allergies—Recommendations from a Multidisciplinary Group of Experts"

_nutrients, 2022, doi:10.3390/nu14194016_

Round 1
Reviewer 1 Report
This paper is a review on the role of nutrition in infancy on the development of atopy in newborn infants. It is a well written and concise review.
I have three remarks.
1. The the review deals mainly with the infant with risk factors, while also infants without a risk factor can develop atopy. A large study done in Europe showed that giving prebiotics to low risk infants reduced the incidence of atopy to a level seen in breast fed infants. Why did you not include non high risk infants in your review and recommendations? See Piemontese PlosOne 2011 and Gruber J Allergy Clin Immunology 2010
2. Although you write that there is no real argument to start in high risk infants in the first week of life with -semi-elementary feeding ininfants when breastmilk is available, you still advise to do so. Why?
3. Basically you follow published advises of international organisations, so what is the new message in this paper?
Author Response
We sincerely acknowledge all the constructive criticism and helpful suggestions. Please find the answers to all your remarks below. All modifications made to the manuscript are highlighted by the MS Word track changes system.
This paper is a review on the role of nutrition in infancy on the development of atopy in newborn infants. It is a well written and concise review.
I have three remarks.
- The review deals mainly with the infant with risk factors, while also infants without a risk factor can develop atopy. A large study done in Europe showed that giving prebiotics to low risk infants reduced the incidence of atopy to a level seen in breast fed infants. Why did you not include non high risk infants in your review and recommendations? See Piemontese PlosOne 2011 and Gruber J Allergy Clin Immunology 2010
We do appreciate that the discussion of allergy development in low- or non-risk infants is necessary and certainly worth of attention. However, and in this particular manuscript, our aim was to address the prevention of atopy in children with an increased risk, by nutritional manipulation in infancy through the choice of formula. We strongly believe the benefit associated to prevention strategies is higher for the population at risk, and therefore actions directed at this population will have a lower cost/benefit ratio than those directed at the whole population. Of note, one does expect that if a certain product reduces the risk in the more susceptible group it will also provide benefit to the non-at-risk babies. Still, the exploration of such relationship falls outside the scope of our article.
Piemontese, PlosOne 2011. This manuscript reports the tolerance and safety of a specific formula containing a mixture of oligosaccharides in early infancy. Although an important one, it falls outside the scope of our manuscript, as explained above.
Gruber J Allergy Clin Immunology 2010. This study addresses the potential benefit of using a formula with specific prebiotic and immunoactive oligosaccharides on the occurrence of atopic dermatitis in the first year of life. Although important, the results pertain the outcomes observed in healthy, non-at-risk babies (those with allergic relatives were excluded), and therefore falls outside the scope of our manuscript, as explained above.
- Although you write that there is no real argument to start in high risk infants in the first week of life with -semi-elementary feeding in infants when breastmilk is available, you still advise to do so. Why?
Indeed, as shown in the footnote of the summary of recommendations, there is limited evidence and a formal recommendation cannot be provided. We state that – “When exclusive breastfeeding in not an option, intact protein cow’s milk-based formula should be used to feed or supplement healthy infants irrespectively of their allergy risk”. We then add that “without disregarding the previous point, short-term bottle feeding of otherwise breastfed babies up to one week of age should be avoided (as breastmilk is usually sufficient); however, and when needed, adequate supplementary options include donor breast milk, eHF or AAF.” The later sentence is not a formal recommendation, but a reminder that, if absolutely necessary, acceptable alternative options do exist for the short-term feeding of very young babies, although neither of them is strongly advisable or supported by the evidence.
- Basically you follow published advises of international organisations, so what is the new message in this paper?
We agree that this paper does not bring new evidence; instead, it intends to be a repository of the recent evidence from extensive reviews, meta-analysis and guidelines in a field where there is still a wide variation of practices across countries. The use of expensive formulas with protein manipulations that deviate from the gold standard is an option that should derive from reasonably solid evidence, something that is currently lacking in what concerns the pH formulas. Therefore, we consider that re-visiting the topic and providing some recommendations is appropriate in order standardize clinical practice and bring it closer to the scientific evidence.

Reviewer 2 Report
This article focuses on one opinion that infant formula has a role in reducing the development of allergic disease in non-breastfed infants and unlike previous beliefs that hydrolysed formula is more appropriate, this article proposes the use of intact protein formula. It also proposes a role for the addition of synbiotics in the prevention of allergic disease.
Firstly, this article is logical and it presents a point of view worth discussing. Also, at the end of the article, it gives some suggestions, which are of great value for feeding infants.
More notably, the authors point out that some health care professionals continue to use pHF and eHF even though they are no longer recommended, which shows that the authors are more familiar with the clinic.
However , there are some areas that need improvement.
Firstly, there are few citations in the article to support the idea that intact proteins can reduce the incidence of allergy.
Secondly, there is a lack of specificity regarding whether there are adverse effects of using intact protein in high-risk infants such as preterm infants.
Finally, the level of evidence cited in the article is not high enough, indicating that this issue is still controversial. It is hoped that more studies and experiments can be conducted to obtain a definite result.
Author Response
Note to Reviewers:
We sincerely acknowledge all the constructive criticism and helpful suggestions. Please find the answers to
all your remarks below. All modifications made to the manuscript are highlighted by the MS Word track
changes system.
This article focuses on one opinion that infant formula has a role in reducing the development of allergic
disease in non-breastfed infants and unlike previous beliefs that hydrolysed formula is more appropriate,
this article proposes the use of intact protein formula. It also proposes a role for the addition of synbiotics
in the prevention of allergic disease.
Firstly, this article is logical and it presents a point of view worth discussing. Also, at the end of the article,
it gives some suggestions, which are of great value for feeding infants.
More notably, the authors point out that some health care professionals continue to use pHF and eHF even
though they are no longer recommended, which shows that the authors are more familiar with the clinic.
However , there are some areas that need improvement.
Firstly, there are few citations in the article to support the idea that intact proteins can reduce the incidence
of allergy.
We do agree that the number of publications supporting the protective role of intact proteins for allergy
development is limited. However, we would like to emphasise that this should be viewed as a secondary
output: our main aim was to explore the role of pHF and eHF in allergy prevention, and that was duly
supported by several citations. In fact, the evidence shows that pHF and eHF have no preventive role in
atopy development; as a consequence, the use of intact protein formulas is a natural choice. The discussion
on the potential protective role of intact proteins comes at the end of the section, more as a complement
to the previously-discussed topic: “In this scenario, one may wonder whether the ingestion of intact cow’s
milk formula (vs. pHF and eHF) early in life may actually promote oral tolerance to cow’s milk”. The number
of citations is limited, but so is the evidence in this issue, as we acknowledge later on the text: “Of note,
the mentioned studies are observational, and rigorously-designed clinical studies are needed to fully explore
this association and to demonstrate causality”.
Besides, and importantly, we did not aim at performing an extensive review of all available evidence. In
fact, and as we also highlight in the text, the results of many of the individual studies on formulas are only
applicable to the particular composition used each study (“[…] however, it has been suggested that the
qualitative changes suffered by the peptides and the hydrolysis’ method - rather than just the degree of
hydrolysis – impact the preventive potential of each formula [29]. As a consequence, findings for a particular
formula should not be generalized to other formulas”). Therefore, and while we certainly acknowledge the
importance of each individual study, we have chosen to base our opinion article on meta-analysis, revisions
and guidelines, which have gathered together the results of many individual studies and approach them
from a global perspective.
Secondly, there is a lack of specificity regarding whether there are adverse effects of using intact protein in
high-risk infants such as preterm infants.
We agree that preterm babies may present a special group of infants with particular needs in terms of
formula composition, particularly in what concerns caloric intake. However, we are unaware of studies
specifically addressing the role of the formula composition in the allergy risk of this (or other) highly
vulnerable group, or describing potential adverse effects of cow’s milk intact protein during routine feeding
of non-allergic babies. Further evidence is needed to provide guidance.
Finally, the level of evidence cited in the article is not high enough, indicating that this issue is still
controversial. It is hoped that more studies and experiments can be conducted to obtain a definite result.
We absolutely agree with you that the level of evidence at this point is below what would be desirable.
Indeed, we acknowledge that throughout the manuscript, and particularly in the conclusions: “Importantly,
there are still major knowledge gaps in what concerns nutrition effects on allergy prevention: all
recommendations described above are based on limited or very limited evidence, and further studies are of
utmost importance to confirm (or contradict) our vision in this field. Given the continuously-increasing
allergy incidence and its consequences in terms of children development, growth and quality of life, a huge
effort should be placed in the identification of effective preventive mechanisms. We therefore urge infant
formula manufacturers and the scientific community to gather their efforts in order to fully comprehend the
mechanisms underlying allergy development, so that effective nutritional strategies can be established
(particularly during the critical first 1000 days).” Still, decisions are being made every day in clinical practice:
paediatricians, nurses and other healthcare professionals are called to give their opinion on the best
formula to feed high-risk babies, and their counselling is not always in line with the latest evidence (as
scarce as it might be). We therefore felt the need to revisit this topic and issue a number of
recommendations in the field, in order to standardize clinical practice and bring it closer to the scientific
evidence, but by no means implying that our recommendations are robust: on the contrary, we highlighted
that they are based on limited evidence and that more studies based on stringent methodology are needed
to substantiate or improve the currently available evidence.

Round 2
Reviewer 1 Report
Te authors must make more clear that this review only deals with infants at risk to develop allergic symptoms. It should me added to the title and made clear in the beginning of the abstract and introduction. Now, only haf-way of the introduction high risk is mentionned.
Secondly, the authors seem to assume that the development of allergy is only due to dysbiosis. They advise to consider a formula with synbiotics for infants at risk for dysbiosis. Given the complex course of the development of allergy, should in my opinion be written "consider the use of a formula with pro, pre or synbiotics in infants at risk to develop allergy.
Finally, I still can not agree with the use of eHF or AAF formula in the first week of life when fomula is not or not enough available. There is simply no evidence for that advise. And, these formula's are very expensive.
Author Response
Note to Reviewers:
We sincerely acknowledge all the constructive criticism and helpful suggestions. Please find the answers to all your remarks below. All modifications made to the manuscript are highlighted by the MS Word track changes system.
The authors must make more clear that this review only deals with infants at risk to develop allergic symptoms. It should be added to the title and made clear in the beginning of the abstract and introduction. Now, only half-way of the introduction high risk is mentionned.
We have included “infants at risk of developing allergies” to the title, to the abstract, to the introduction and to the expert comment.
Secondly, the authors seem to assume that the development of allergy is only due to dysbiosis. They advise to consider a formula with synbiotics for infants at risk for dysbiosis. Given the complex course of the development of allergy, should in my opinion be written "consider the use of a formula with pro, pre or synbiotics in infants at risk to develop allergy”.
We have included this in the expert comment in line 464 and in box 1 of the manuscript.
Finally, I still can not agree with the use of eHF or AAF formula in the first week of life when formula is not or not enough available. There is simply no evidence for that advise. And, these formula's are very expensive.
We have included a new text in line 455 of the manuscript and in box 1.
